# Computing Policies That Account for the Effects of Human Uncertainty During Execution in Markov Decision Processes

### Abstract

When humans are given a policy to execute, there can be policy execution errors and deviations in policy if there is uncertainty in identifying a state. This can happen due to the human agent's cognitive limitations and/or perceptual errors. So an algorithm that computes a policy for a human to execute ought to consider these effects in its computations. An optimal Markov Decision Process (MDP) policy that is poorly executed (because of a human agent) maybe much worse than another policy that is suboptimal in the MDP, but considers the human-agent's execution behavior. In this paper we consider two problems that arise from state uncertainty; these are erroneous state-inference, and extra-sensing actions that a person might take as a result of their uncertainty. We present an approach to model the human agent's behavior with respect to state uncertainty, which can then be used to compute MDP policies that accounts for these problems. This is followed by a hill climbing algorithm to search for good policies given our model of the human agent. We also present a branch and bound algorithm which can find the optimal policy for such problems. We show experimental results in a Gridworld domain, and warehouse-worker domain. Finally, we present human-subject studies that support our human model assumptions.

## 1 NOTES

## 2 Introduction

Markov Decision Processes (MDPs) have been used extensively to model settings in many applications((Boucherie and Van Dijk 2017),(Hu and Yue 2007),(White 1993)) but when the agent that has to act in such a scenario is a human, we need to consider how the execution changes. The human maybe the agent executing the policy because (for example) certain actions cannot be done by the robots (legal or ability limitations) or the human has to step-in due to a failure in automation (fail safe). Human performance in the latter is called out-of-the-loop (OOTL) performance (Endsley 2017). For such cases, computing a good human policy is important, and to do so one ought to consider human limitations and behavior when computing a policy for people to execute.

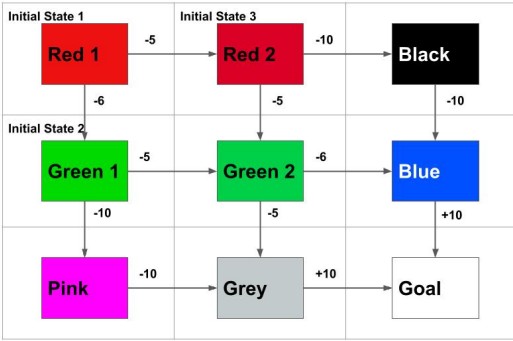

Figure 1: A colored gridworld domain in which an agent determines the states by the color; initial states are annotated as well.

In this work we focus on problems during execution by a human agent that arise from uncertainty during state inference. Specifically, we focus on two issues; erroneous state inference, and extra-sensing behavior(policy) of the human agent due to uncertainty. Uncertainty can arise from perceptual limitations or cognitive limitations. When a person is uncertain about the current state, we may take additional sensing actions, or take longer to decide and may still end up deciding incorrectly. A simple example of this is if we are lost or unsure of our location when traveling in a new place, we might repeat sensing actions (looking for signposts or landmarks) to locate ourselves better.

For our work, we consider that the human (when uncertain) may take additional sensing actions which may or may not be helpful. We treat these actions as coming from an extra-sensing policy of the human, as they are not part of the state-action mapping in the policy given to the human to execute; they are a consequence of the human's uncertainty. We cannot programmatically control these, as we would with a non-human agent. In this work, we represent the extra-sensing policy as a unique extra-sensing action per state (we will discuss this more).

As a result of the extra-sensing action, the agent's uncertainty might get reduced, or the additional sensing may not make a difference in correcting errors (expected/average outcomes may not improve); a human agent might do

them anyway as these additional sensing actions are their only options to resolve uncertainty. In our work, we are *not* trying to remove uncertainty during execution; we expect it to persist in certain problems despite efforts to minimize it. We instead try to account for the effects of uncertainty when computing a policy to get better policies. An optimal policy (in terms of domain dynamics) which ignores the effects of uncertainty on execution behavior can be markedly suboptimal. To help build intuition, we present an example using a "colored-gridworld" domain in Figure 1.

In this simple example, an agent determines the current state by the color of its current position (imagine a person is inside a room, and each room is a grid position). The colors of grid positions include two shades of red, and two shades of green in the grid, and lets assume the human can get confused between similar shades of colors. Then the optimal policy that ignores state identification errors and cost of extra-sensing actions can be worse off than a simpler policy that has the same action for similar states; in this example the better policy would map both red states to the down action, and both green states to the "right" action when getting to the goal state. The costs might seem more than the optimal policy, but if we account for errors (like confusing the two red shades) and cost of extra-sensing action to resolve uncertainty, the optimal policy would be of lower value than the other; we show this in later in our human subject experiments.

With respect to the likelihood of the human taking an extra-sensing action, it can be affected by not only the domain states, but also the human agent's mental state. Mental modeling a human agent can be challenging. However, we do not need a complex mental model for this problem. Instead, we use the likelihoods of events related to the human's behavior; specifically, for a given state what are the possible states for them (how are they uncertain), and what is the most likely state. We also model how they take extra-sensing actions. Using a probabilistic model of the human agent's inference and behavior, we can compute better policies for human execution by accounting for how people make mistakes and respond to uncertainty.

In this paper, we formally define the problem of computing a policy that accounts for human behavior under uncertainty in an MDP. We do this by converting it into a constrained Partially Observable Markov Decision Process (POMDP). Our contributions include : (1)A model of human agent behavior under state uncertainty (2) A Hill Climbing algorithm and branch-and-bound algorithm to find reactive controller policies that incorporate our human model; reactive controllers map observations to actions in POMDPs. (3) We validate our approach using experiments on a Gridworld domain. (4)Lastly, we present the results of a human subject study which shows evidence for the effects of uncertainty that we assume and model in this work. We delay the related work comparison to the end as the exposition of the idea first helps us better compare to the related work.

## 3 Human Model And Problem Definition

In this work, we assume that the underlying MDP is fully observable (the ground truth state is knowable), and an AI agent can detect the state and execute the optimal policy perfectly. However, when the human enacts a policy, the human's limitations leads to suboptimal execution due to state uncertainty during execution. This uncertainty turns the MDP into a POMDP where the observation random variable is over states; we will elucidate this further in Section 3.2. The problem this paper addresses is how to compute a policy that accounts for the human's state-uncertainty and behavior due to this uncertainty. This paper will build up to the formal definition of the problem by first discussing the human-agent model used. Then we will define how it is incorporated into a POMDP for computing a better policy for humans.

### 3.1 Human Model

The human model is defined using the probability of inference events and extra-sensing events given a ground-truth state. For a set of states $S$, we define the human model as $H = \langle p_c, p_u, \psi_0, \psi_1 \rangle$, and the terms are defined as follows:

- $p_c : S \times S \to [0, 1]$ $p_c$ is the probability of classifying (identifying) one state as another; $p_c(\hat{s}|s^*)$ where $s^*$ is the true state, and $\hat{s}$ is the best-guess state that the human agent thinks it is. We will use the $\hat{s}$ symbol above a state to indicate the human's guess of the state.

- $p_u : S \times \{S_i \in 2^s\} \to [0, 1]$ is the probability of being uncertain over a set of states ($S_i$) for a given true state. For example, $p_u(\{s_i, s_j\}|s^*)$ is the probability of the human considering $\{s_i, s_j\}$ as the possible states when the true state is $s^*$. We will refer to such $S_i$ sets as a "possible-set".

- $\psi_0 : S \to [0, 1]$ determines the probability of the extra-sensing action being taken when the human infers only one possible state but is not certain, or was not able to infer any state.

- $\psi_1 : S \to [0, 1]$ determines the probability of the extra-sensing action when the agent is uncertain about the right policy action. This happens when the human considers two or more states as possible, and the policy conflicts between these two states. If there is more pressure to act, or the human-agent is impulsive, the probability of extra-sensing action would be lower, and so $\psi_1$ would be lower as well.

The arguments of $p_c(.)$ and $p_u(.)$ can be seen as capturing the mental-state or belief state of the human agent for a given ground truth state $s^*$; $p_c(.)$ will give the likelihood of a state being the most likely state in the human's mind, and $p_u(.)$ will give the likelihood of a set of states being the possible states in the human's mind.

### 3.2 POMDP With Human Execution Under Uncertainty

Given this human model, the problem of computing a reactive controller for the POMDP due to Human Uncertainty in Execution (POMDP-HUE) is defined by the tuple $\langle S, A, T, r, \gamma, p_i, H, S_2 \rangle$. Each of the terms are defined as follows:

- $S$ is the set of states in the MDP.

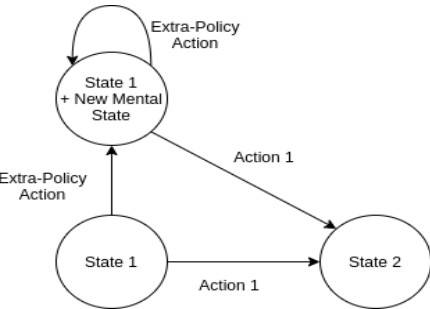

Figure 2: Additional state added to MDP for State 1 to account for different inference likelihoods by the human agent

- $A$ is the set of actions in the domain, and an additional $a^+$ which is the extra-sensing action.

- $S_2$ contains one successor state to every state in $S$, and is reached only when $a^+$ is taken. This captures the change in human's mental-state (beliefs about the state) after the extra-sensing action. This is illustrated for a single state in Figure 2. Using $S_2$ folds the human's mental state into the state space. This is needed because of the extra-sensing action the human takes due to uncertainty.

- $T : \{S \cup S_2\} \times A \times \{S \cup S_2\} \to [0, 1]$ is the transition function that outputs the likelihood of transition from one state to a successor state after an action. This includes the extra-sensing action ($a^+$) dynamics. States in $S$ and $S_2$ have the same action dynamics except for the extra-sensing action $a^+$, as illustrated in Figure 2.

- $r : \{S \cup S_2\} \times A \to \mathbb{R}$ is the reward function. This includes the cost/reward associated to extra-sensing actions.

- $\gamma$ is the discount factor

- $p_i : S \to [0, 1]$ is the probability of a state being the initial state

- $H$ refers to the human model as defined by $\langle p_c, p_u, \psi_1, \psi_0 \rangle$. These terms are probability functions defined for states in $S \cup S_2$. We discuss how to compute these terms in section 4.

We consider that the human's inference does not continually get better by repeating $a^+$ actions, hence the self loop from the state in $S_2$ in Figure 2. There can be different successor mental states for different extra-sensing actions or policies taken from the same state. For this presentation, we limit ourselves by abstracting the human's extra-sensing behavior in each state to an action $a^+$ and it's associated new state which will have the human's (expected) updated beliefs.

The **objective** in the POMDP-HUE problem is to output a *deterministic* policy ($\pi_d : S \to A$) for the human agent, which optimizes for policy value (equation 3) after accounting for effects of uncertainty –that we will shortly formalize– determined by the human model $H$. The deterministic policy is a mapping from a problem state to an action. The state is what is inferred by the human, and can be seen as a noisy observation emitted by the real state. This

makes it a POMDP, and why we refer to the objective as computing a *reactive controller* for a POMDP.

One might argue for defining the policy over the combination of the most likely state and the set of possible states, i.e. the arguments to function $p_c(.)$ and $p_u(.)$. However, there are two negative consequences: (1) We make the policy larger for the human to memorize which can add to the likelihood of misremembering policy actions between similar belief states; (2) we still do not remove the effects of uncertainty; it only shifts how the person is uncertain. If (for example) the policy is different between two belief states that differ only in the most likely state, and the human is not sure which the most likely state is, the uncertainty can still result in extra-sensing actions. In some problems it maybe worth giving the policy to the human over such a belief-space (combining the arguments of $p_c(.)$ and $p_u(.)$), but we note that it can still result in extra-sensing actions, and also carries the cost of a larger policy size. Here we limit the policy to be over the state-space, which is the same as the arguments of $p_c(.)$.

Let us return to reactive controllers; these were defined in prior works (Littman 1994) (Meuleau et al. 2013). A reactive controller for a POMDP implies a control policy based on the current state's observation only; this maps to the inferred state in the human's mind for our problem. As one might surmise, we make the assumption that the human's inference and uncertainty is influenced by the current state only. This limitation can be important in OOTL (out-of-the-loop) performance when the human has to step in due to automation failure; automation tends to make people complacent and not paying much attention (Endsley 2017) and so we might only expect current state information to be accessible to the human. Additionally, high-stress –such as in OOTL or high speed manufacturing settings– can hamper cognitive function on tasks(Sandi 2013). So expecting the human agent to track history accurately, and make good inferences based on that is a tall order; limiting policy conditioning to the current state (using a reactive policy) can be safer and kinder to the person in such settings.

Other than the policy being conditioned on the current state, we use the following consideration: If a person is uncertain over a set of possible states in their mind, and if the policy conflicts between these states, then the human is more likely to take extra-sensing actions to try to resolve uncertainty (which we show in our human subject studies). Put another way, the uncertainty that matters when executing a policy, is about what the right action is; it is not to perfectly detect the state. If the action is the same across possible states then there is no need for additional resolution. In accordance with this, we will shortly define the likelihood of the extra-sensing action ($a^+$) as a function over the policy.

Since extra-sensing actions and uncertainty affect execution of a policy, any deterministic policy given to the human ($\pi_d$), when actually executed by the human becomes a stochastic policy; $\pi_p : (S \times A|\pi_d) \to [0, 1]$. The first effect from uncertainty, is the likelihood of taking an extra-sensing action in a state ($s* \in S \bigcup S_2$) is defined as follows:

$$\pi_p(s*, a^+|\pi_d) = \psi_0(s*) + (1 - \psi_0(s*)) \times \psi_1(s*) \times$$
$$\sum_{S_i \in 2^S} p_u(S_i|s*) \times \mathbb{1}[0 < \sum_{s_1, s_2 \in S_i} \mathbb{1}[\pi_d(s_1) \neq \pi_d(s_2)]] \quad (1)$$

where $\mathbb{1}[.]$ is the indicator function. Succinctly, the equation says is that if one of the states in the possible-set $(S_i)$ has a policy action that doesn't match with another, then the likelihood of extra-sensing action $(a^+)$ can increase proportional to $p_u(S_i|s*)$(the probability of inferring the possible-set $S_i$). For any given state, the number of possible-sets with non-zero probability are likely to be few, and not the full powerset $(2^S)$. For example, in a gridworld setting, the possible-sets for the current state –determined by the agent's position– may involve neighboring states (positions); however, the agent is unlikely to think that it could be much further away.

The aforementioned extra-sensing action could translate to many types of actions in a problem instance; these could include calling a supervisor or colleague for help, or re-checking state features. The specific dynamics of the extra-sensing actions are domain dependent. For this paper's presentation, we limit the effect of extra-sensing actions in that it can improve the inference of the human agent, but doesn't change the ground truth state. In terms of the problem definition, this means the POMDP-HUE state transitions from a state in $S$ to the corresponding state in $S2$ like in Figure 2.

In addition to taking extra-sensing actions, the other effect of uncertainty is on the likelihood of choosing a policy action from the given policy $\pi_d$, and is as follows:

$$\pi_p(s*, a|\pi_d) = (1 - \pi_p(s*, a^+, \pi_d)) \times$$
$$\sum_{s_i \in S} p_c(\hat{s}_i|s) * \mathbb{1}[\pi_d(s_i) = a] \quad (2)$$

This means that the likelihood of an action is the sum of the likelihood of it's associated states being inferred as the current state. This is multiplied by the probability of not taking the extra-sensing action in the state, which ensures the probabilities sum to 1.

Finally, we define the overall value of the original policy $\pi_d$ given to the human as follows:

$$V(\pi_d) = \sum_{s \in S} p_i(s) * V_{\pi_p}(s) \quad (3)$$

where $V_{\pi_p}(s)$ is the state value in the input MDP by following the stochastic policy $\pi_p$ which included the effects of uncertainty. The value is a weighted sum of state value, where the weights are the initial state likelihood given in $p_i(.)$.

## 4 Computing Human Model Parameters

Our work focuses on the computing the policy for a human model defined in terms of $\langle p_c, p_u, \psi_0, \psi_1 \rangle$. Part of the appeal of this approach for us in modeling the human agent, is that we only need probabilities of events. We do not have to make any assumptions on inference-limitations such as bounded or noisy-rational assumption on human inference (Simon 1990), (Zhi-Xuan et al. 2020). The probabilities can be estimated from empirical data. We present an empirical approach to collect the data needed to compute the probabilities; we use this approach in our human subject studies as well.

Since the model parameters are dependent on the state, we can collect data by testing the human agent on just the task of state detection. The human agent is presented multiple instances (trials) of each state –state samples are ordered randomly– and asked to look at the state following a predefined perception-policy. The perception policy could be as simple as a time limit on looking at state information before acting; for example, in manufacturing, a time limit is important as it translates to cost. Alternatively, the perception policy could be a predefined series of perception actions. After the perception-policy, we ask what they think the possible states are (can be more than one) and what they think the most-likely state is. Most importantly, we ask if they would like to confirm their answer of they think the correct state is, or look at the state again (extra-sensing) before confirming their answer. For this process, we would count the following for each state:

- $C_i(\hat{s}_i|s^*)$:The number of times a state $(s_i)$ was inferred (most-likely state to the human) for a given state $(s^*)$.

- $C_u(S_i|s^*)$: The number of times the person inferred a set of possible states $(S_i \subset 2^S)$ for a given state. This includes the empty set, and singleton sets with one state. The count $C_u(\{s_i\}|s^*)$ of a singleton set $\{s_i\}$ will be atleast as much as $C_i(\hat{s}_i|s^*)$; it can be greater if (for one of the trials) the human only considers one state as possible but is also uncertain, and takes an extra-sensing action.

- $C_{e0}(s^*)$: How often a person took an extra-sensing action when uncertain, *and* their set of possible states was either 1 or none.

- $C_{e1}(s^*)$: How often a person took the extra-sensing action *when* they reported they were uncertain over two or more possible states.

Note than when a person takes one or more extra-sensing actions, we consider all subsequent counts separately; these counts are used is to compute the human model parameters for states in the set $S_2$, separate from $S$.

Using the data collected we compute the human model parameters as:

$$p_c(\hat{s}_i|s^*) = \frac{C_i(\hat{s}_i|s^*)}{\sum_{s_j \in S} C_i(s_j|s^*)} \quad (4)$$

$$p_u(S_i|s^*) = \frac{C_u(S_i|s^*)}{\sum_{S_j \in 2^S} C_u(S_j|s^*)} \quad (5)$$

$$\psi_0(s^*) = \frac{C_{e0}(s^*)}{\sum_{S_j \in \{S:S \in 2^S, |S| \leq 1\}} C_u(S_j|s^*)} \quad (6)$$

$$\psi_1(s^*) = \frac{C_{e1}(s^*)}{\sum_{S_j \in \{S:S \in 2^S, |S| > 1\}} C_u(S_j|s^*)} \quad (7)$$

These definitions are so that $\psi_0$ captures the likelihood of taking extra-sensing action even without any inference conflict (but the human was not confident in their inference), or the human was unable to infer any state. On the other hand, $\psi_1$ is the likelihood of taking extra-sensing actions when uncertain; this covers the cases when the human's inference results in 2 or more states being possible (conflicting inference). If $\psi_1$ is much less than 1, it would mean that the human decides to act more often than resolve uncertainty even if uncertain. One can think of $\psi_1$ as a reflection of the pressure to act on the human agent, or a reflection of their patience to resolve uncertainty.

# 5 Policy Computation For POMDP-HUE

Finding an optimal solution to the POMDP-HUE problem is at least as difficult as computing a reactive (memoryless) controller for a POMDP, which is what our problem reduces to if one ignores the extra-sensing action; this can be done by setting $\psi_0 = 0, \psi_1 = 0$ for all states. Computing a reactive controller has been shown to be NP-hard ((Littman 1994)). To handle this computational complexity, we present two algorithms. One is a hill-climbing algorithm for computing good albeit suboptimal policies quickly, and to handle larger state spaces. The other is a branch-and-bound algorithm for computing the optimal policy at higher computational cost, which is suitable for smaller state spaces and also for bounding the suboptimality of the hill-climbing approach for larger state spaces.

## 5.1 Human-Agent Policy Iteration(HAPI)

We call our hill climbing approach Human-Agent Policy Iteration (HAPI) which takes the greedy best step to change the policy while accounting for human agent's uncertainty effects. In HAPI we start with a random deterministic policy ($\pi_d$), and compute the corresponding stochastic policy after state aliasing ($\pi_p$ as defined by equations 1 and 2). We then determine the value of this stochastic policy by equation 3. Then (in the hill climbing step) for each possible policy change we compute the new policy value, and select the action to change the policy. This is repeated until no better changes can be made. Each step's computational complexity is $O(|S|^4|A|)$; this is because each step tests a number of changes no more than $|S||A|$, and the value of a fixed policy can be computed in $O(|S|^3)$ by computing the state transition likelihoods for that policy and using the following closed form computation (standard equation for value computation in a Markov Reward Process (MRP) (See (Ibe 2013) for more details on Markov processes):

$$\vec{v_s} = (I - \gamma * P_{ss'})^{-1} * \vec{r_s} \tag{8}$$

Where $\vec{v_s}$ is the vector of state values, $P_{ss'}$ is the transition probability matrix for a given policy, and $\vec{r_s}$ is the vector of expected rewards at each state (which can be computed for a fixed policy).

The total time taken for HAPI will naturally be problem specific; the number of improvement steps will depend on the initial point and the possible improvements in the domain. Additional random restarts can improve the outcome, as is common in hill-climbing approaches.

## 5.2 HUE Branch-And-Bound Policy Search (H-B&B)

HAPI is helpful to quickly find a good policy. However, if one wanted the optimal policy, then the following branch and bound approach –which we will refer to as H-B&B can be used for smaller state spaces. It can also be used to bound the suboptimality of the policy found by HAPI, which can be used to decide if further iterations of HAPI would be worthwhile or not.

This branch-and-bound searches in policy space by choosing an action for a state at each level in the search tree. We assume the reader is familiar with the basics of branch and bound (Brusco, Stahl et al. 2005). At any given point in the policy search, only a partial policy is defined. We need a lowerbound, and an upperbound to determine if the node in the search tree should be expanded. We set the initial lowerbound as the value of the policy output by HAPI search.

We still need a helpful upperbound that accounts for the extra-sensing action. To compute this, we use an MDP relaxation of the POMDP-HUE for a given partial policy. This is done by assuming perfect state observability *only* for the remainder of the undefined states (policy not yet assigned), and using a lower-bound for the likelihood of errors and extra-sensing actions for the other states. We call this a "Partially-Controlled MDP" (PC-MDP). We compute the optimal policy (including extra-sensing actions) for this PC-MDP using value iteration and that is the upperbound. This idea of using an easier MDP to bound the state-value in branch-and-bound is similar to the bound employed in (Meuleau et al. 2013) except theirs does not consider or allow any notion of extra-actions. The gist of it is as follows: If one can set a lower-bound for the probability of state-misidentification and extra-sensing actions in all states, then by optimizing for the remainder of the policy action probability in each state, the policy-value obtained will be equal to or greater than any other possible policy completion. A trivial lower-bound would be to assign zero probability to errors, i.e. $s_1 \neq s_2 \rightarrow p_c(s_1|s_2) = 0$, and extra-sensing actions ($\pi_p(s, a^+|\pi_d) = 0$) for the states whose policy is not yet defined. Then optimizing the PC-MDP policy would give the upperbound for state-value. Our bound considers the effect of prior decisions in H-B&B to give a tighter, more helpful upperbound for the search process. This is done by using the human model $H$ parameters and lower-bounding the likelihood of extra-sensing action by removing undefined states from the probability computation in Equation 1. The pruning effects of our upperbound will be shown in the results. If the reader is interested in the details and proof of the bound, please see our supplemental material [1].

In each step in the branch and bound, we need to run value iteration on the PC-MDP. In our algorithm, we stop value iteration after a certain number of iterations; we set number of iterations(k) to 1000 in our experiments. We then take an upperbound for each state's value computed as $v_k(s) +$

---

[1] https://tinyurl.com/5n9y7dzh

$\frac{\epsilon * \gamma}{1-\gamma}$ (Chapter 17 (Russell and Norvig 2021)) where $\epsilon$ here is $||v_k - v_{k-1}||$. This error is added to the policy value to set the upperbound.

The size of the policy search tree is unfortunately large; it is $|A|^{|S|}$ if we assume the same number of actions ($|A|$) in each state. However, a good upperbound and ordering the states intelligently can greatly prune the tree. We order the nodes in the search tree using the following score:

$$score(s) = (\frac{1}{|S|} + p_i(s)) \times \sum_{s' in S} p_c(s|s') \times \max_{a \in A(s')} r(s', a)$$

(9)

This function increases the score of a state based on how likely a state is to be the initial state ($p_i(s)$) since those state values determine the overall policy value (Equation 3). It also considers the likelihood that other states are confused with it, because the policy decision for those states will affect the state value for others too (due to state confusion). This likelihood is scaled by the max reward possible in the other states. The scaling is because we want to order policy decisions for states based on how much they influence the policy-value; so we prioritize decisions affecting higher reward/lower cost states. This score (Equation 9) can help us make pivotal policy decisions sooner in the search process, and work with the upperbound to prune the search tree faster.

## 6  Experiments and Results

We tested our algorithms on two qualitatively different domains; gridworld and warehouse-worker. We present the gridworld experimental results here as we thought it most informative for the ideas in this work and evaluating performance. In the warehouse-worker domain, an agent has to make packing decisions for a set of products to be shipped to a customer. The errors come from not knowing which is the best box for a set of products (small, medium, or large boxes). A detailed description of the warehouse-worker domain and experimental results are in the supplemental material [1].

For testing our algorithms, we repeated HAPI ten times (10 random restarts) for each experimental setting and consider the best value as the output from HAPI. As for H-B&B search, it ofcourse need only be run once. All results can be consistently reproduced from our codebase all variability in the program is controlled by a random-seed parameter that is set to 0.

### 6.1  Experimental Setup

First we present the Gridworld experiments on $5 \times 5$ grids where H-B&B was allowed to run to completion. We varied the properties of the MDP to see how well HAPI performs compared to H-B&B. The actions for each state (defined by agent position) are the standard ones; these are move up, down, left, and right. The goal is the bottom right square like in Figure 1, albeit *without* the colors (colors are used in the human subject studies). The goal state is an absorbing state, and the reward is 100 upon transitioning into it.

When the agent takes the extra-sensing action ($a^+$), it reduces by half both the error likelihood ($p_c()$ of incorrect state detection) and probability of incorrect possible-sets $p_u$ (possible-sets that have states other than the ground truth). Taking $a^+$ results in a state transition to a parallel state in $S_2$ (as in problem definition and Figure 2). $S_2$ states have the same action effects but with different $p_c$ and $p_u$ functions. Subsequent extra-sensing actions from this state, returns to the same state. This means additional extra-sensing actions from $S_2$ does not change the inference outcomes ($p_c$ and $p_u$) of the agent in our experiments.

All action transitions are stochastic with a 5% chance of transitioning to a random neighboring grid position or stay in place. All actions will have a random cost for each experimental setting. An invalid action (like moving up from the top of the grid) results in the agent staying in the same state and incur the cost assigned to that state and action. The extra-sensing action has a cost of 1.

As for the likelihoods of confusing states ($p_c$), we define the likelihood of confusing a grid state (position) with another based on the L1 distance between positions. $p_c$ is defined in Equation 10. The equation is simply saying that neighboring states are much more likely to be confused with each other than with those further away.

$$p_c(s|s') = \frac{1/(L1(s, s') + \mathbb{1}[s = s'])^m}{\sum_{s'' \in S} 1/(L1(s, s'') + \mathbb{1}[s = s''])^m} \quad (10)$$

where $m$ is a scalar. We set it to 5 for our experiments. This makes the likelihood of confusing one state with another that is more than 1 step away to be very small. Lastly, we add the $+1$ to avoid dividing by zero. As for the possible-sets in $p_u$, we limit ourselves to sets of size 1 and 2. The probability of each set ($p_u(.)$) are computed using equation 11.

$$p_u(\{s_2, s_1\}|s*) = p_c(\hat{s_2}|s_1) * p_c(\hat{s_1}|s*) + \\ p_c(\hat{s_1}|s_2) * p_c(\hat{s_2}|s*) \quad (11)$$

Note $s_1$ can be the same as $s_2$ in Equation 11; those cases correspond to the $p_u(.)$ probability for possible-sets of size 1. Lastly, in the agent model, $\psi_0$ was set to 0.05 –which means when there is no inference conflict the agent may still take $a^+$ 5% of the time– and $\psi_1$ to 0.9.

With respect to the experimental settings, we first present results of a 5x5 grid, with one additional mental-state ($S_2$ state) per state. We used a smaller grid first because while HAPI (hill-climbing) can handle larger sizes of grids, branch and bound (H-B&B) speed drops very quickly. This is because the policy space grows as $|A|^{|S|}$; even a 5x5 grid has $4^{25} \approx 1.1 \times 10^{15}$ policies. However, the search process eliminates most policies quickly, and the bound helps immensely with pruning the policy space. Engineering improvements to speed up H-B&B through parallelization and memory management is left as future improvements. We posit that for a single task's policy (for a human agent) even state-sizes around 25 can be sufficient for some problems. For example, a basic car-maintenance policy for owners would have a few states and associated actions to deal with issues such as

oil-change, car battery-health and such. In the supplemental section[1], we discuss an application motivated domain-example via the warehouse-worker domain that can be described with 12 states. If the problem requires a larger state-space, we can use HAPI search to find policies.

To evaluate the algorithms, we focused on varying three parameters: (1) The discount factor $\gamma$, whose default value is 0.7; (2) the likelihood of random actions whose default value is $\rho = 0.05$; (3) A "reward noise range" parameter (RNR) to add random rewards to each of the actions in the grid and whose default value is 2; an $RNR = 2$ would result in random rewards for each action in the range $[-1, 1]$, i.e., uniformly distributed about 0.

We chose to vary the discount factor since a larger discount factor couples the policy decisions more strongly (since the state value is affected by states further away). We also chose to add random rewards to each of the actions other than the goal actions to make the search more challenging. Lastly, increased random action likelihood meant that states which had both high reward and high cost actions are less attractive than if there was no random action likelihood.

Our code was implemented in python using "pybnb" library for branch and bound, and PyTorch and NumPy for matrix operations. The experiments were run on a PC with Intel® Core™ i7-6700 CPU, running at 3.40GHz on Ubuntu 20.04 with 32 GB of memory.

## 6.2 Results

We first present the values of the policies discovered for the 5x5 grid experiments in Table 1. The best policy discovered by HAPI approach was either very close to optimal, or optimal in all cases. The takeaway is that for this experimental setting, HAPI with 10 iterations found either a very competitive value or the optimal policy value in all cases. We found this to be the case in the 10x10 grid setting as well (with the random costs and stochastic transitions) in Table 2. For those experiments, we only used H-B&B to find an upperbound and use it to define the suboptimality of the policy value found by HAPI; the last number in each table entry is the ratio with the upperbound. In the 10x10 grid setting as well, HAPI was able to perform well. A related point is that H-B&B was able to find a good upperbound to the policy value within 30 minutes, which can be helpful in deciding if further iterations of HAPI might be worthwhile or not. In our supplemental material[1], we present results for the time-taken for the same experiments, as well as the number of nodes openend by H-B&B search. The nodes opened is a fraction of the policy space which shows the efficacy of the upperbound used in the search; even in the worst result (most nodes opened) it is a miniscule fraction of the total (ratio of $4.2 \times 10^{-21}$)

## 6.3 Human Subject Experiments

For our human subject experiments we wanted to see if our assumptions hold with respect to human policy execution under state uncertainty. We use a small grid world setting, and use colors to define each state in order to introduce state uncertainty from perception. The full grid is as illustrated in Figure 1. In the underlying MDP, each action has costs

| Discount Factor (RNR=2, $\rho$=0.05) | | Reward Noise Range ($\rho$=0.05, $\gamma$=0.7) | | Random Action Probability (RNR=2, $\gamma$=0.7) | |
|---|---|---|---|---|---|
| $\gamma$ | Values | RNR | Values | $\rho$ | Values |
| 0.3 | 11.87, 11.87, 1.0 | 0 | 33.67, 33.67, 1.0 | 0.05 | 34.16, 34.17, 1.0 |
| 0.5 | 19.05, 19.05, 1.0 | 1 | 33.89, 33.89, 1.0 | 0.1 | 33.65, 33.65, 1.0 |
| 0.7 | 34.16, 34.17, 1.0 | 2 | 34.16, 34.17, 1.0 | 0.15 | 33.08, 33.08, 1.0 |
| 0.9 | 69.45, 69.45, 1.0 | 4 | 34.75, 34.75, 1.0 | 0.2 | 32.46, 32.46, 1.0 |

Table 1: Policy value results for a 5x5 grid. Each of the primary columns changes one experiment-parameter, and holds the other two constant. Each entry has the best policy value from HAPI, the optimal value found by H-B&B, and the ratio of the two. All values rounded down to two decimal places

| Discount Factor (RNR=2, $\rho$=0.05) | | Reward Noise Range ($\rho$=0.05, $\gamma$=0.7) | | Random Action Probability (RNR=2, $\gamma$=0.7) | |
|---|---|---|---|---|---|
| $\gamma$ | Values | RNR | Values | $\rho$ | Values |
| 0.3 | 3.17, 3.48, 0.91 | 0 | 11.14, 11.47, 0.97 | 0.05 | 11.56, 12.29, 0.94 |
| 0.5 | 5.26, 5.65, 0.93 | 1 | 11.27, 11.83, 0.95 | 0.1 | 11.17, 11.86, 0.94 |
| 0.7 | 11.56, 12.29, 0.94 | 2 | 11.56, 12.29, 0.94 | 0.15 | 10.77, 11.4, 0.94 |
| 0.9 | 43.9, 45.91, 0.96 | 4 | 12.28, 13.48, 0.91 | 0.2 | 10.36, 10.97, 0.94 |

Table 2: Policy value results for a 10x10 grid. Each of the primary columns changes one experiment-parameter, and holds the other two constant. Each entry has the best policy value from HAPI, the best upperbound found by H-B&B after 30 minutes, and the ratio of the two. All values rounded down to two decimal places

as illustrated in Figure 1, and all undisplayed actions have a cost of -10. There is a reward for reaching the goal at the bottom right position (+10). After reaching the goal position, the state then changes to a random new position from a set of initial states.

Our objective in this study was to first build an averaged human model using the procedure described in Section 4, and then use that model to compute a policy that accounts for the uncertainty effects. The performance using this policy is then compared to the human agents' performance using the optimal policy for the underlying MDP. The state space was designed so that some sets of states (the color associated to them) are visually similar, like "Green 1" and "Green 2" in Figure 1, and cause uncertainty.

All participants were recruited using the "Prolific" service for online studies, and prescreened using using their service for vision (can see colors clearly). We also asked 3 questions at the start of our study to test if participants could distinguish between lighter and darker shades that look similar. All prolific participants are above 18 years of age, and equal division of male and female participants (as they identify) was requested for the study (Prolific handles this part). No other demographic information was collected. Gender or age based comparisons are out of the scope of this study. Our human-subject studies had IRB approval, and we gave clear information about the purpose of the experiments to our participants in the consent form before the experiments. We also debriefed the participants, and gave the option to contact us for more information.

In the first phase, we collect data to build an (averaged)

model of a human agent for the task. This means computing $< p_c, p_u, \psi_0, \psi_1 >$ for each state (including $S_2$ states). We do this using a preliminary study that displays the colors used in the main study, and asks the human to match a color displayed (colored square) on the left of the screen to a list of numbered colors on the right of the screen. The color was only shown for 0.5 seconds, but the table on the right was permanently displayed. They were given the option to see-again if they were uncertain by pressing the back arrow key; doing so gives us a clear indication of the extra-sensing action.

In this study, before the human can submit their answer or ask to see the color again, we ask them to enter their guesses as to which colors they think it could have been. Example, if a color $Green1$ was displayed the user might enter $(1, 2)$ which are the indices for the two green shades, or just enter one color if they were confident in their decision.

The participants were paid a flat amount, as well as an additional 0.1 dollars for every correct answer as an incentive to get it correct. We used data from 16 participants for this phase, each participants was asked 15 questions; colors were randomly sampled during testing. We use the data collected to compute the parameters of the human model as per the equations in Section 4.

As one might expect, the two shades of green, and two shades of red causes participants to request to see the color again, as well as make the most mistakes. All the unique (non-similar) colors that we tested the participants with were easily identified with almost no errors and no extra-sensing actions. We saw this for many unique colors during our initial testing, and so felt confident that we were not showing the colors for too short a duration. One interesting note was that people seemed more likely to confuse the darker shade of red with the lighter shade than vice versa; this was not the case for the shades of green which was much less confusing.

After computing the human model parameters, we use it to compute the optimal policy using H-B&B for the grid MDP in the study. We also computed the optimal policy by value iteration which ignores state-uncertainty. These were the two policies given to the humans in phase 2 of the study, and are displayed in Figure 3; the policy on the left is the optimal policy of the MDP (ignores) uncertainty, and the policy on the right is the one that accounts for uncertainty. The discount factor $\gamma$ was set to 0.9 to compute the policies.

Just as in phase 1, we display a color on the left of the screen for 0.5 seconds. This color corresponds to their current position in the grid as in Figure 1. We ask the participant to press the arrow key corresponding to the color seen using the policy displayed on the right, and so navigate through the underlying grid. After each step the color of the new position is displayed. The participant can see a color again by pressing the "Control" button (extra-sensing $a^+$ action). We track every button press and their progress through the grid. When they reach the goal state, the state is reset to another of the initial states, for a total of 3 times. The initial states are shown in Figure 1. We used 20 participants for this phase, and had to drop one data point as the data suggested they were randomly guessing for both policies.

Using the data collected, we wanted to see if using the

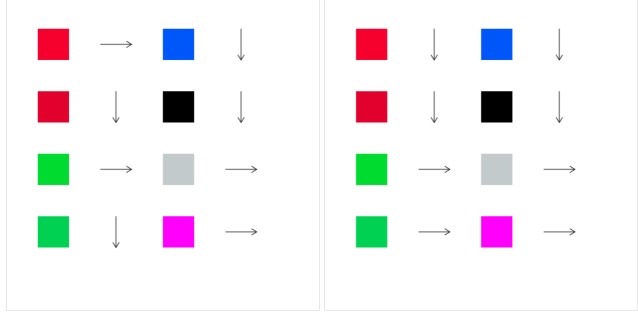

Figure 3: The two policies for the second phase of human subject experiments. The left policy is the optimal policy *without* considering the effects of uncertainty, and the right is the optimal policy after accounting for uncertainty.

policy that accounted for uncertainty translated to more reward accrued. We took the difference between the reward accrued in each run (initial to goal state) with it's corresponding run in the other policy. On average the cost incurred by the policy that accounts for uncertainty is less by 1.45. We ran a dependent t-test for paired samples; the same participant did both corresponding runs, and so we treat the data as paired samples. We set the significance level for t-test at $\alpha = 0.05$, and ran the paired t-test using the scipy-stats library in python (Virtanen et al. 2020) to see if the policy with uncertainty was better (one-sided test). We got a t-test statistic value of 2.240 which corresponds to a p-value of 0.0147. Thus we can reject the null hypothesis that the policy *with* uncertainty gives the same or worse value than the optimal policy without uncertainty.

For our experiments, given the simplicity of the problem we found it sufficient and easier to build an averaged human model and use it for all participants. Ideally one would build a human model unique to a person. Such an averaged model could be tuned with fewer additional data points per person, using a Bayesian approach to computing the parameters (using a dirichlet distribution for tracking the priors). Overall, our human-subject studies give support to the idea that policies which consider state uncertainty are executed faster and more reliably.

## 7   Related Work

If the effect of uncertainty was limited to only erroneous state detection (as captured by $p_c(.)$ in the human model), one can frame the problem in this paper as computing a reactive controller for a POMDP and use prior methods in (Littman 1994) and (Meuleau et al. 2013). However, none of the prior methods handle the case where additional (extra-sensing) actions are taken by the agent due to uncertainty. This can result in different policies between our approach and prior reactive controller approaches. We verified this by setting $\psi_1, \psi_0$ to be zero (so no extra-policy actions are taken); doing so changes our H-B&B algorithm to compute the optimal POMDP reactive controller based on $p_c(.)$ as the observation likelihood. We found that this policy and it's value was suboptimal in gridworld experiments when there

were higher costs for extra-sensing actions; this fits our expectations as one would ignore extra-sensing actions when computing a standard reactive controller for a POMDP.

An extension and generalization of the reactive controllers work is computing history-based controllers (Kumar and Zilberstein 2015) which considers mapping a history of observations to actions as opposed to just the current observation (which is a state-history of 1). For this work we limited ourselves to history of 1 state. One could reasonably extend our approach to longer history-based controllers if needed. There are additional concerns such as the higher cognitive load for inferring the most likely sequence of states, and larger policy size that must be considered when doing so.

In the direction of considering human errors, there is work that considers "blindspots" in an agent's representation, and how to transfer control between an automated agent and a human based on their blindspots (Ramakrishnan et al. 2019). These blindspots can arise due to a mismatch in the state space during training versus execution, or limitations in representational capabilities. There is a follow-up work that focuses specifically on a human agent's blindspots (Ramakrishnan et al. 2021) and reducing errors. Our work attacks the problem from a different angle. Instead of minimizing errors for a given policy, we try to compute a policy that accounts for human errors and human's behavior in response to uncertainty.

## 8 Conclusion and Future Work

In this paper, we define the problem of computing a reactive policy that accounts for human execution behavior under state uncertainty. We formalized a probabilistic model of the human agent's inference and behavior, as well as how to compute the parameters in it. We then presented two algorithms (HAPI and H-B&B) to compute policies for our problem, and show experimental results in a gridworld setting. We also have results for a warehouse-worker domain in the supplemental material[1]. Lastly, we conducted human subject studies to show an example of how the human model can be empirically derived, and use it with our H-B&B algorithm to compute the optimal policy for our problem. We show that this policy resulted in statistically more reward accrued than the optimal MDP policy that ignores the effects of uncertainty. Our human-subject studies supports the considerations we make for our human model such as expecting identification errors between similar states and extra-policy actions.

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
