# OpenReview forum: "Computing Policies That Account for the Effects of Human Uncertainty During Execution in Markov Decision Processes"
_icaps-conference.org/ICAPS/2022/Workshop/XAIP — XAIP 2022_

### Official Review · Reviewer_W3jm · 2022-04-25
**Accounting for human execution errors in planning**

**Rating:** 5
**Confidence:** 5

**Review:**

This paper presents an approach to account for human execution errors in the planning phase. Two types of errors are considered: (1) incorrect state inference; and (2) performing extra information gathering actions to determine the current state. In both cases, the human diverts from the prescribed policy. The paper presents an approach to account for such execution uncertainties in the planning process. The problem is formulated as a POMDP-HUE, which includes a model of human agent performance. The human model is formulated by considering their likelihood of misclassifying a state, being uncertain between a subset of states, probability of executing extra sensory actions to resolve uncertainty over a state, and a bias term that affects the probability of executing the sensory action. The parameters for human model are determined by collecting data about human performance. Given a model of the human agent, the paper presents a hill climbing approach and a branch and bound algorithm to compute reactive controller policies to solve this problem.  Empirical results are provided on a grid-world domain and on a warehouse domain, along with user study results to support their modeling assumptions about the human agent.

While the paper’s contributions are sufficient for a workshop paper, the relevance to XAIP workshop is not clear. I was looking for at least a description that highlights how legible actions will aid or when explanations can be useful in this context, even though the contributions are not in core XAIP.  I'm rating the paper as a weak reject because the connections to this workshop have not been clearly established.

The authors claim that the human model parameters can be computed by empirical data. This is a costly process and may not be possible to generalize the gathered information.

Minor comments: “Section 1: Notes” has no content. It should be removed.

---

### Official Review · Reviewer_Ytmq · 2022-04-27
**Well written paper with clear ideas**

**Rating:** 7
**Confidence:** 4

**Review:**

Summary:
This paper presents a modification to the MDP formulation to model scenarios where humans are executing policies. This model can take into account human uncertainty in the execution of the policy, by modeling their uncertainty over states and their likelihood of taking extra sensing actions to gain more information. The authors present a Hill-climbing algorithm (HAPI) and a Branch-and-Bound (H-BNB) algorithm to solve such POMDP-HUEs to create policies that can account for human uncertainty (e.g. by having similar policies for states that might confuse a human). They show empirical experimental results and results from a human subject experiment to back their formulation.

Review:
The paper is well written, and the exposition provides sufficient clarity to the reader. The examples and discussion are convincing and motivate the problem well.

Following are some minor issues with the content of the paper:
1. There are a lot of references to the supplement, so it may be worth condensing some of the exposition in the paper and including additional information from the supplement.
2. The experimental results provided are for only one domain, with only two problem instances. It would be interesting to see how this formulation scales to other MDP settings. Specifically, including runtimes of the various methods would help better distinguish them.
3. The applicability of this paper to the XAIP workshop can be made more clear. While the paper deals with using mental models of humans and interacting with them in a planning context, it is not very clear how this fits into the umbrella of "explainability".

Typos and edits:
Remove empty section (1. NOTES)
"these counts are used is to compute" - remove "is" (Section 4)

---

### Meta-Review · Program_Chairs · 2022-04-30

**Recommendation:** Accept
**Confidence:** 4

**Metareview:**

Both reviewers note that the paper presents a novel and sound approach to addressing the problem of generating policies that could account for possible mistakes people may make. While both reviewers were happy with the work in itself, there were some concerns about how the work exactly connects to the workshop. However, this could be an issue that could be fixed by adding an extended discussion along the lines that the reviewer suggested. So I am recommending it be accepted with the expectation that the authors would add the required discussion.

---

### Decision · Program_Chairs · 2022-04-30

Accept